# Efficacy of Tocilizumab Therapy in Different Subtypes of COVID-19 Cytokine Storm Syndrome

**DOI:** 10.3390/v13061067

**Published:** 2021-06-03

**Authors:** Oleksandr Oliynyk, Wojciech Barg, Anna Slifirczyk, Yanina Oliynyk, Vitaliy Gurianov, Marta Rorat

**Affiliations:** 1Department of Anaesthesiology and Intensive Care, Bogomolets National Medical University, 01601 Kyiv, Ukraine; alexanderoliynyk8@gmail.com; 2Department of Emergency Medicine, Pope John II State School of Higher Education in Biala Podlaska, 21-500 Biala Podlaska, Poland; aslifirczyk1@gmail.com; 3Department of Internal Medicine, Pneumonology and Allergology, Wroclaw Medical University, 50-367 Wroclaw, Poland; wojciech.barg@umed.wroc.pl; 4Department of Immunology and Allergology, Bogomolets National Medical University, 01601 Kyiv, Ukraine; janinaoliynyk@gmail.com; 5Department of Medical Statistics, Bogomolets National Medical University, 01601 Kyiv, Ukraine; i_@ukr.net; 6Department of Forensic Medicine, Wroclaw Medical University, 50-367 Wroclaw, Poland

**Keywords:** monoclonal antibodies, ARDS, cytokine storm syndrome, inflammation

## Abstract

Background: Cytokine storm in COVID-19 is heterogenous. There are at least three subtypes: cytokine release syndrome (CRS), macrophage activation syndrome (MAS), and sepsis. Methods: A retrospective study comprising 276 patients with SARS-CoV-2 pneumonia. All patients were tested for ferritin, interleukin-6, D-Dimer, fibrinogen, calcitonin, and C-reactive protein. According to the diagnostic criteria, three groups of patients with different subtypes of cytokine storm syndrome were identified: MAS, CRS or sepsis. In the MAS and CRS groups, treatment results were assessed depending on whether or not tocilizumab was used. Results: MAS was diagnosed in 9.1% of the patients examined, CRS in 81.8%, and sepsis in 9.1%. Median serum ferritin in patients with MAS was significantly higher (5894 vs. 984 vs. 957 ng/mL, *p* < 0.001) than in those with CRS or sepsis. Hypofibrinogenemia and pancytopenia were also observed in MAS patients. In CRS patients, a higher mortality rate was observed among those who received tocilizumab, 21 vs. 10 patients (*p* = 0.043), RR = 2.1 (95% CI 1.0–4.3). In MAS patients, tocilizumab decreased the mortality, 13 vs. 6 patients (*p* = 0.013), RR = 0.50 (95% CI 0.25–0.99). Conclusions: Tocilizumab therapy in patients with COVID-19 and CRS was associated with increased mortality, while in MAS patients, it contributed to reduced mortality.

## 1. Introduction

The coronavirus disease 2019 (COVID-19) pandemic means there is a lot of focus on the immunopathology problems caused by this disease. Many studies discussing an altered immune status in COVID-19 mention a cytokine storm syndrome (CSS) [1]. The activation of macrophages by severe acute respiratory syndrome coronavirus-2 (SARS-CoV-2) infection, which starts in the lungs, is a primary source of pro-inflammatory cytokines. The development and exacerbation of the inflammatory process, which consequently leads to lung failure and other organ dysfunction, is the result of a dysregulated macrophage response [2,3]. Diagnosis of CSS is based on three criteria: elevated circulating cytokine levels, acute systemic inflammatory symptoms, and secondary organ failure (mostly pulmonary, renal, and hepatic) due to hyperinflammation beyond a normal response to the pathogen [1]. Only a handful of researchers indicate that CSS is heterogenic and comprises subtypes. This includes macrophage activation syndrome (MAS), cytokine release syndrome (CRS), and sepsis [4]. Despite similar clinical symptoms, these subtypes differ in their pathomechanism and may consequently require different therapeutic options [5,6]. Some researchers suggest that improper qualification in the treatment of these subtypes may result in an increased mortality rate [5].

One of the treatment options studied in patients with CSS in the course of COVID-19 is based on the use of anti-interleukin-6 receptor monoclonal antibodies from immunoglobulin subclass IgG1, among which tocilizumab is the most commonly used [7,8]. The effectiveness of tocilizumab has been demonstrated in cytokine storm disorders, including *hemophagocytic lymphohistiocytosis (HLH)*, idiopathic multicentric Castleman disease, and chimeric antigen receptor (CAR) T-cell–induced cytokine storm [9]. The role of interleukin 6 (IL-6) in controlling viral infections has been proven in influenza A, herpesvirus, and SARS-CoV-1 infections [10]. As the increased level of IL-6 also correlates with COVID-19 severity and mortality, researchers hypothesise that blocking cytokine signaling may assist in the clearance of the SARS-CoV-2 virus [1,11].

Tocilizumab is mainly prescribed to patients with severe COVID-19 and is reported to have several significant side effects. Opinions about its usefulness in COVID-19 are ambiguous and the majority of researchers consider more studies on the feasibility and effectiveness of this drug are required [8,12,13]. The discrepancies between study results may arise from heterogeneous individual responses, which may depend on the specific pathogenic weight of IL-6 in the complex scenario of the SARS-CoV-2-induced cytokine storm [14]. The purpose of this research was to study the outcome of treatment with tocilizumab in different CSS subtypes.

## 2. Materials and Methods

In a retrospective study, we analysed the medical records of 276 consecutive patients with severe COVID-19, hospitalised from 01.02.2020 to 01.11.2020 in an infectious diseases intensive care unit at Kyiv City Clinical Hospital № 4.

Inclusion criteria comprised:Confirmed SARS-CoV-2 infection (positive RT PCR test, *reverse transcription polymerase chain reaction*);Presence of bilateral interstitial pneumonia on a computed tomography (CT) scan;Respiratory failure with arterial partial pressure of oxygen < 60 mm Hg on room air.

Exclusion criteria comprised: serious comorbidities that could potentially affect the course of the disease (cardiogenic pulmonary oedema, pulmonary embolism, recent brain stroke, advanced chronic pulmonary diseases, malignancies, diabetic ketoacidosis, decompensated chronic kidney or liver diseases, and autoimmune diseases), pregnancy, and participation in other clinical studies.

Patients were split into three groups: patients with MAS (n = 28, 10.1%), with sepsis (n = 24, 8.7%) and with CRS (n = 224, 81.2%).

The diagnostic criteria of the Histiocyte Society (HLH-2004 criteria) [15] were used to diagnose MAS. According to this definition, the patient must meet at least five out of eight of the following criteria:1.Fever over 38.5 °C for more than 7 days;2.Splenomegaly;3.Cytopenia (affecting ≥2 of 3 peripheral blood lineages):▪Haemoglobin (<9 g/dL).▪Platelets (<100 × 10^9^/L).▪Neutrophils (<1.0 × 10^9^/L);4.Hypertriglyceridemia (>3 mmol/L);5.Hypofibrinogenemia (<1.5 g/L);6.Haemophagocytosis in bone marrow or spleen or lymph nodes;7.Ferritin >500 µg/L;8.Elevated IL-6.

In our group, criterion 6 was not taken into account—patients did not have a bone marrow or lymph node biopsy.

Sepsis was diagnosed according to the commonly used definition of sepsis [16]: Sepsis is life-threatening organ dysfunction caused by a dysregulated host response to infection. Organ dysfunction is identified as an acute change in total sequential organ failure assessment (SOFA) score ≥2 points consequent to the infection. Additionally, patients with procalcitonin (PCT) ≥2.0 ng/mL were included in that group.

CRS was diagnosed in patients who had clinical symptoms (especially hyperthermia, general weakness, and myalgia), together with elevated laboratory markers (C-reactive protein (CRP), ferritin, and IL-6) of severe inflammation but had procalcitonin <0.2 ng/mL [17,18].

The medical report comprised the patient’s medical history, physical examination, abdominal ultrasound examination result and laboratory tests results, performed on admission: complete blood count, PaO_2_/FiO_2_ (the ratio of arterial oxygen partial pressure to fractional inspired oxygen), C-reactive protein, procalcitonin, fibrinogen, D-dimer, IL-6, and ferritin. Respiratory index PaO_2_/FiO_2_ was determined according to the method of P. Marino, using a BGA 101 gas analyzer made by Wondfo, Guangzhou, China. C-reactive protein was determined by enzyme immunoassay using a Humastar 600 analyzer (Wiesbaden, Germany) and a reagent kit HUMAN (GmbH, Wiesbaden, Germany). Procalcitonin was determined by the enzyme immunoassay using an immunological VIDAS analyzer with accessories 4700023 (bioMérieux’s Marcy l’Etoile France); kit for the enzyme immunoassay of procalcitonin in human plasma and serum BRAHMS VIDAS^®^ BRAHMS. The quantitative content of fibrinogen was determined using the Clauss method using a Siemens BFT II semi-automatic two-channel coagulometer (Siemens Healthcare Diagnostics, Schwalbach am Taunus, Germany) and a Multifibren U reagent kit (Marburg, Germany). To determine the D-dimer, an enzyme-linked immunosorbent assay was used: Humastar 600 analyzer (Human GmbH, Wiesbaden, Germany) and a HUMAN reagent kit. To determine interleukin-6, a set of reagents for the enzyme-linked immunosorbent assay for the determination of the concentration of interleukin-6 in blood serum and urine Human Interleukin 6 was used, IL-6 ELISA KIT (DRG Diagnostics GmbH, Marburg, Germany). The determination was carried out on an automatic ELISA analyzer EVOLIS Twin Plus (Bio-Rad, Marnes-la-Coquette, France). Ferritin was determined by enzyme immunoassay using a Humastar 600 analyzer (Human GmbH, Wiesbaden, Germany) and a reagent kit HUMAN (Human GmbH, Wiesbaden, Germany).

Treatment with tocilizumab is not recommended in COVID-19 patients with an increased level of procalcitonin [19]. Consequently, this approach was offered to all patients with CSS and MAS, but not to patients suspected of having sepsis. Only 55/252 (21.8%) patients gave written, informed consent to the treatment. Treatment with tocilizumab was started between day 8 and day 14 from the onset of symptoms. Tocilizumab was administered intravenously at a dose of 400 mg for two consecutive days. All patients were treated with dexamethasone 6 mg i.v. daily, low-molecular-weight heparin administered subcutaneously in prophylactic doses, antibiotics (in case of suspected or confirmed bacterial infection), and balanced fluid therapy.

### Statistical Analysis

MedCalc^®^ Statistical Software version 19.5.6 (MedCalc Software Ltd., Ostend, Belgium; https://www.medcalc.org; 2020) was used for the statistical analysis. To present quantitative data, the median of the indicator (Me) and the interquartile range (QI–QIII) were calculated; the distribution differed from the normal one by the Shapiro–Wilk criterion. Frequency (%) was calculated for qualitative indices. For comparison of quantitative features between groups, the Kruskal–Wallis criterion was used; posterior comparisons were made according to the Dann criterion. For comparison of qualitative features in more than two groups, the chi-square criterion was used, posterior comparisons were made by the Fisher exact test with the Bonferroni correction taken into account. Fisher’s exact test was used to compare frequencies in two groups. To quantify the clinical effect, the risk ratio (RR) and its 95% confidence interval (95% CI) were calculated. To quantitatively assess the degree of influence of factor signs on the risk of a fatal outcome, the method of building and analysing logistic regression models was used. The impact of the factors were measured by the value of the odds ratio (OR), for which 95% CI was calculated. For all statistical tests, the *p* value < 0.05 was considered significant.

## 3. Results

Table 1 presents the medical data of the groups studied. There were no statistically significant differences in age and sex between the groups. In our patients, CRS was the most common (224/276, 81.2%) CSS subtype and seems to be the mildest of the three subtypes considered. Compared to the MAS and sepsis subtypes, clinical symptoms in CRS were less pronounced, with lower body temperature and without hepato- or splenomegaly. All 38/276 (13.8%) patients who were not diagnosed with acute respiratory distress syndrome (ARDS) had CRS. Laboratory parameters were also less pronounced, with no leukocytopenia and relatively mild thrombocytopenia, hyperferritinemia, and interleukinemia (Table 1).

In contrast, patients with MAS and sepsis demonstrated a more severe course of COVID-19. The median body temperature was 38.1 °C and 39.2 °C in sepsis and MAS, respectively. Hepato- and splenomegaly were observed in MAS patients. All patients with sepsis or MAS had ARDS. We also observed significant differences in laboratory tests between the groups studied (Table 1). Most importantly, mortality in CRS patients (14.4%) was significantly lower (*p* < 0.001) as compared to MAS and sepsis (67.9% and 50%, respectively). There was no difference in mortality between MAS and sepsis groups (*p* = 0.778).

Figure 1 presents the differences in clinical effects of tocilizumab on the course of MAS and CRS. In MAS patients, tocilizumab improved the course of COVID-19 with respect to mortality (RR = 0.46 (95% CI 0.25–0.86)) and the need for intubation (RR = 0.57 (95% CI 0.36–0.90)), while, in patients with CRS, it increased the risk of both death (RR = 2.1 (95% CI 1.1–4.2)) and intubation (RR = 1.9 (95% CI 1.1–3.4)).

Tocilizumab increased the risk of death (OR = 2.8 (95% CI 1.1–6.8)) and the risk of intubation (OR = 2.6 (95% CI 1.1–6.0)) in the CRS patients after adjustment (by variable logistic regression) by the PaO_2_/FiO_2_ index and ARDS. In turn, in the MAS patients, it decreased the risk of death (OR = 0.01 (95% CI 0.001–0.22)) after adjustment by the PaO_2_/FiO_2_ index.

The total number of CRS patients with ARDS was 186 (36 treated with tocilizumab). The risk of death for the patients treated with tocilizumab was 27.8% and, without the treatment, 13.3%, RR = 2.1 (95% CI 1.1–4.1).

An in-depth analysis of features that may be associated with the risk of intubation or death was conducted. Single-factor and multi-factor logistic regression models were used for analysis. Table 2 and Table 3 show the results of a single-factor analysis and Table 4 shows the results of a multifactor logistic regression.

A single-factor logistic regression analysis was also carried out for all patients with ARDS and treated with tocilizumab. There were no statistically significant differences between intubated vs. non-intubated and deceased vs. survived for any investigated laboratory parameters.

To identify a pool of independent risk factors of intubation and death, a set of features was selected in multi-factor logistic regression models (stepwise method, parameters with *p* < 0.1 were included in the analysis). Due to the small groups, no separate analysis was performed for the CRS and MAS groups.

From all the biomarkers studied (Table 2) multivariate logistic regression analysis revealed few independent risk factors of intubation: leucocyte count, fibrinogen and D-dimer concentration; death: ferritin concentration, leucocytes, and lymphocyte count (Table 4).

## 4. Discussion

A severe course of COVID-19 is generally combined with one of the CSS subtypes, usually CRS, but the most severe cases are usually complicated by sepsis or MAS. All CSS subtypes are characterized by life-threatening hyperinflammation which supports a cytokine storm and ultimately leads to multiple organ failure [20]. In the spectrum of cytokines involved in the pathogenesis of CSS, IL-6 and ferritin are of great importance [21].

The role of IL-6 in the immunopathogenesis of COVID-19 is supported by extensive research data showing an increase in the concentration of this cytokine in patients’ serum. This is proposed for monitoring the severity of COVID-19 [11,22]. According to Otsuka and Seino [23], in patients (n = 1302) with severe COVID-19, IL-6 was three times higher than in patients with a mild or moderate course (*p* < 0.001). Its concentration was associated with bilateral lung damage (*p* = 0.001) and fever (*p* = 0.001). Other studies show that elevated IL-6 is correlated with the progression of ARDS (*p* = 0.03), the requirement for mechanical ventilation, and mortality risk [11,24]. According to Ruan et al. [25], the average concentration of IL-6 (11.4 ± 8.5 ng/mL) in deceased patients is significantly higher (*p* < 0.001) than that of survivors (6.8 ± 3.6 ng/mL). In our study, IL-6 concentrations differ statistically significantly within the investigated groups, although the difference in absolute values between MAS and CRS is small (64 v 60 pg/mL, *p* < 0.001) (Table 1).

Ferritin concentration is increased in all subtypes of CSS but, in MAS, it rises sharply and may exceed the normal value even by a factor of several hundred [26]. The highest ferritin level in our patients with MAS was as high as 129,000 ng/mL, and the average value was 5894 ng/mL. According to the HLH-2004 recommendations [15], the characteristic feature of MAS is an increase in ferritin levels above 500 ng/mL, but some authors [23] suggest that the average ferritin level in MAS exceeds 3000 ng/mL. In contrast, ferritin levels in sepsis patients are much lower and, in our study, the median was 957 ng/mL. These data are consistent with the results of Giamarellos-Bourboulis EJ et al. [26], in which COVID-19 and sepsis patients had a mean ferritin level of 954 ng/mL (508.4–5394). Thus, ferritin seems to be the most distinctive marker for MAS. In addition, in our study ferritin concentration is found to be an independent risk factor for death.

Monoclonal antibodies against the IL-6 receptor, including tocilizumab and sarilumab, have a pronounced immunosuppressive effect and have been used in the treatment of COVID-19. There is a general opinion that tocilizumab may be effective in severe forms of COVID-19. It is considered that such therapy is appropriate and can reduce the mortality rate and the need to switch patients to forced ventilation with intubation. However, data on the effectiveness of treatment with tocilizumab in COVID-19 patients are ambiguous [7,13,27,28,29,30,31]. Two multicenter trials addressed this issue, and published recently in NEJM, gave contradicting conclusions. Rosas et al. [30] compared clinical outcomes in 294 patients treated with tocilizumab added to standard therapy with 144 patients on standard therapy only (placebo group). The authors found that the use of tocilizumab did not improve clinical status or decrease mortality at day 28. In contrast, Gordon et al. [31] presented convincing data that tocilizumab significantly improved clinical outcomes and survival in 353 critically ill COVID-19 patients, who received tocilizumab, as compared to 402 placeboes. It is difficult to unequivocally assess the reasons for this discrepancy. However, it should be noted that the efficacy of tocilizumab was demonstrated in critically ill patients hospitalized in ICU, while in the group where efficacy was not revealed, only about half were hospitalized in ICU. Therefore, one of the potential hypotheses assumes that tocilizumab is more effective in more severe forms of COVID-19. Thus, reports of failure of tocilizumab therapy should be taken with caution, although this could, to some extent, be explained by the heterogeneity of the examined COVID-19 populations.

The most important finding in our research is the demonstration that the clinical effect of tocilizumab is significantly different in the CSS subtypes investigated. In MAS patients, treatment with tocilizumab reduced mortality and the need for intubation by approximately half. In contrast, in the CRS group, the use of tocilizumab almost doubled the risk of intubation and death. MAS appears to be fairly rare and its prevalence in our study was 10.1%. It is believed that the presence of MAS in COVID-19 patients worsens the disease course and results in a higher mortality rate. In classic autoimmune MAS, mortality can be up to 100% if not treated appropriately [4]. Thus, only timely diagnosis and early treatment (including tocilizumab) can reduce mortality in patients with COVID-19 and MAS.

Our study revealed a higher median body temperature, more frequent presence of hepato- and splenomegaly, severe hyperferritinemia, hypofibrinogenemia, and pancytopenia, i.e., anaemia, leukopenia, and lymphocytopenia and higher D-dimers in patients with MAS. This agrees with findings in MAS resulting from other causes, mostly from rheumatoid diseases [32,33,34]. As it might be difficult to differentiate MAS from sepsis, especially if the viral infection is a trigger for MAS, it is essential to determine procalcitonin and ferritin levels to distinguish patients with MAS from ones with sepsis. Thus, an appropriate differential diagnosis of MAS seems to be of crucial importance for tocilizumab therapy.

The approach to the prescription of tocilizumab should be differentiated. It is common knowledge that, in the case of COVID-19 and sepsis, it should not be administered [19]. In our opinion, it should be used cautiously in patients with CRS. We observed the development of sepsis in two patients after treatment with this medication (Table 5). The target group for tocilizumab treatment should be patients meeting the MAS criteria, which is consistent with the MAS treatment protocol from the Histiocyte Society [15]. Patients with MAS have an extremely high probability of a fatal disease outcome without the use of tocilizumab.

It is difficult to draw conclusions from the results because of the limited number of observations, especially in MAS patients: only 14 patients receiving and 14 patients not receiving tocilizumab. The availability of such a small number of these patients is explained by the rare occurrence of MAS. Therefore, from our perspective, tocilizumab therapy in the case of COVID-19 should be prescribed selectively and considering the CSS subtype that the patient has developed. Further research is necessary to definitively address the effectiveness and appropriateness of its use.

## 5. Conclusions

In patients with COVID-19 and macrophage activation syndrome, tocilizumab contributes to reduced (*p* = 0.013) mortality, RR = 0.46 (95% CI 0.25–0.86). In contrast, in patients with cytokine release syndrome, it is associated with increased (*p* = 0.043) mortality, RR = 2.1 (95% CI 1.1–4.2).

## Figures and Tables

**Figure 1 viruses-13-01067-f001:**
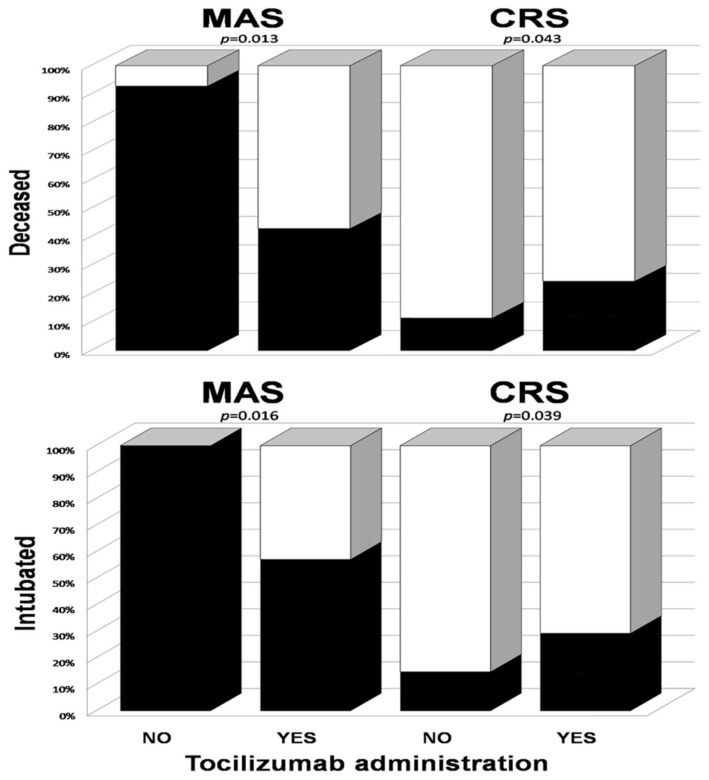
Bar graphs showing the need for intubation and deaths (solid bars) of MAS and CRS depending on the administration of tocilizumab. To compare the risk of an event depending on tocilizumab administration, Fisher’s exact test was used for each group.

**Table 1 viruses-13-01067-t001:** Baseline clinical characteristics and laboratory features of cytokine storm syndrome subtypes, median (QI–QIII).

	CRS (n = 224)	MAS (n = 28)	Sepsis (n = 24)
Age, years	68 (66–71)	69.5 (66–71)	68 (66–71)
Female, (%)	108 (48.2)	13 (46,4)	13 (54.2)
Temperature (on admission), °C	37.5 (37.3–37.8)	39.2 * (39.2–39.2)	38.1 * (38–38.3)
Hepatomegaly, (%)	4 (1.78%)	27 *^,^** (96.4%)	8 (33.3%)
Splenomegaly, (%)	2 (0.89%)	25 *^,^** (89.3%)	3 (12.5%)
General weakness, (%)	212 (94.6%)	26 (92.8%)	23 (95.8%)
Myalgia, (%)	206 (92.0%)	22 (78.6%)	16 * (66.7%)
Ferritin, ng/mL	984 (626.5–1314)	5894 *^,^** (5537–6595)	957 (868–1221)
Interleukin-6, pg/mL	60 (47–72)	64 ** (60.5–66)	95.45 * (88.5–103.45)
Procalcitonin, ng/mL	1.4 (1–1.7)	0.6 ** (0.6–0.7)	5.6 * (3.95–7.1)
Fibrinogen, g/L	2.5 (2.1–3.3)	1.55 *^,^** (1.5–1.6)	2.7 (2.25–3.15)
C-reactive protein, mg/L	52 (48–72)	32 *^,^** (28–44)	79 * (76–97)
D-dimer, ng FEU/mL	1246 (435–1423)	2485.5 * (1978–3115.5)	2005 * (1567.5–2448.5)
Leukocytes, × 10^9^/L	4.2 (3.95–4.3)	1.8 *^,^** (1.7–1.8)	14.6 * (12.3–15.65)
Thrombocytes, × 10^9^/L	126 (102–138)	60 *^,^** (58–66)	87.75 * (83–95)
Lymphocytes, %	24 (22–26)	17 *^,^** (16–18)	23.5 (22–3)
Erythrocytes, × 10^12^/L	3.2 (2.6–3.7)	2.2 *^,^** (2.2–2.3)	3 (2.85–3.15)
PaO_2_/FiO_2_, mm Hg	276 (249–287)	120 * (112–124)	91 * (86–116.5)

Abbreviations: CRS—cytokine release syndrome, MAS—macrophage activation syndrome; Q—quartyl; Laboratory test reference ranges: ferritin 8–143 ng/mL, interleukin-6 <4.0 pg/mL, procalcitonin <0.02 ng/mL, fibrinogen 2.0–4.0 g/L, CRP <5.0 mg/L, D-dimer <500 ng FEU/mL, leukocytes 4.0–9.0 × 10^9^/L, thrombocytes 200–400 × 10^9^/L, lymphocytes 19–37 %, erythrocytes 3.6–4.2 × 10^12^/L, PaO_2_/FiO_2_ (the ratio of arterial oxygen partial pressure to fractional inspired oxygen) 454–495 mm Hg. *: statistically significant difference from the group of patients with cytokine release syndrome, *p* < 0.001. **: statistically significant difference from the group of patients with sepsis, *p* < 0.001.

**Table 2 viruses-13-01067-t002:** Results of a single-factor logistic regression analysis of the risk factors for intubation and death.

	Intubation			Death		
N = 252	OR	95% CI	*p* Value	OR	95% CI	*p* Value
Age, years	6.92	1.20–39.6	0.029	3.41	0.54–21.4	0.188
Temperature (on admission), °C	115.1	20.8–637.8	<0.001	73.6	14.9–363.2	<0.001
Ferritin, ng/mL	5.2	1,31–4,8	0.193	6.01	2.44–14.8	<0.001
Interleukin-6, pg/mL	2.93	1.05–8.2	0.04	2.26	0.76–6.71	0.140
Procalcitonin, ng/mL	0.797	0.333–1.91	0.640	0.721	0.28–1.84	0.491
Fibrinogen, g/L	0.0334	0.00945–0.118	<0.001	0.0567	0.0159–0.202	<0.001
C-reactive protein, mg/L	1.04	0.301–3.57	0.953	0.857	0.225–3.26	0.821
D-Dimer, ng FEU/mL	2.68	1.43–4.99	0.002	2.21	1.14–4.27	0.018
Leukocytes, ×10^9^/L	0.0811	0.0114–0.575	0.012	0.0222	0.00218–0.226	0.001
Thrombocytes, ×10^9^/L	0.00749	0.00148–0.0377	<0.001	0.0082	0.00149–0.0452	<0.001
Lymphocytes, %	0.0034	0.00042–0.0275	<0.001	0.00383	0.00042–0.0347	<0.001
Erythrocytes, ×10^12^/L	0.129	0.0474–0.35	<0.001	0.181	0.0637–0.514	0.001
PaO_2_/FiO_2_, mm Hg	0.00483	0.00091–0.0256	<0.001	0.00743	0.00159–0.0348	<0.001

Abbreviations: PaO_2_/FiO_2_—the ratio of arterial oxygen partial pressure to fractional inspired oxygen; OR—odds ratio; CI—confidence interval.

**Table 3 viruses-13-01067-t003:** Results of a single-factor logistic regression analysis of the risk factors for intubation and death in all patients (MAS + CRS) treated with tocilizumab.

	Intubation			Death		
	Yes (n = 20)	No (n = 35)	*p* Value	Yes (n = 16)	No (n = 39)	*p* Value
Ferritin, ng/mL	1493 (887.5–5927.0)	1056 (668–1534)	0.0950	1377 (1003–6213)	1044 (668–1534)	0.0735
Interleukin-6, pg/mL	67 (60–70.5)	63 (47–72)	0.309	68 (62–72)	63 (47–72)	0.114
CRP, mg/L	54 (45–70.5)	48 (39–72)	0.482	56 (47–72)	48 (39–72)	0.127

**Table 4 viruses-13-01067-t004:** Results of a multivariate logistic regression analysis of the risk factors for intubation and death.

	Intubation, χ^2^_3_ = 55.8 *p* = 0.00000				Death, χ^2^_3_ = 47.4 *p* = 0.00000			
N = 252	Estimate	OR	95% CI	*p* Value	Estimate	OR	95% CI	*p* Value
Ferritin, ng/mL					1.07	2.91	1.00–8.42	0.048
Fibrinogen, g/L	−2.58	0.00446	0.00053–0.0375	<0.001				
D-Dimer, ng FEU/mL	−1.30	0.273	0.0899–0.0153	0.02				
Leukocytes, ×10^9^/L	−0.332	0.0805	0.0153–0.424	0.003	−0.426	0.0392	0.00594–0.259	<0.001
Lymphocytes, %					−0.166	0.0134	0.00085–0.210	0.002

**Table 5 viruses-13-01067-t005:** Adverse reactions after treatment with tocilizumab in CRS and MAS patients.

	MAS (n = 28)	CRS (n = 224)
Adverse Reaction No, (%)		
Neutropenia	1 (7.1%)	2 (4.88%)
Leukocytosis	2 (14.2%)	3 (7.31%)
Thrombocytopenia	2 (14.2%)	3 (7.31%)
Increase ALT	1 (7.1%)	3 (7.31%)
Hypertension	1 (7.1%)	3 (7.31%)
Sepsis	0	2 (4.88%)

Abbreviations: MAS—macrophage activation syndrome, CRS—cytokine release syndrome, ALT—alanine aminotransferase.

## Data Availability

The data presented in this study are available on request from the corresponding author. The data are not publicly available due to ethical aspects.

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
