# Peer review of "Efficacy of Tocilizumab Therapy in Different Subtypes of COVID-19 Cytokine Storm Syndrome"

_viruses, 2021, doi:10.3390/v13061067_

Round 1
Reviewer 1 Report
In this interesting paper the authors have evaluated the effects of tocilizumab in two subtypes of cytokine storm syndrome: the cytokine release syndrome (CRS) and the macrophage activation syndrome (MAS) and found that tocilizumab contributed to reduce mortality in patients affected by MAS but to increase mortality in those affected by CRS.
MAJOR COMMENTS
- Abstract, line 22: In each group, treatment results were assessed depending on whether or not tocilizumab was used. This sentence is not correct. In fact Tocilizumab was only evaluated in cytokine release syndrome (CRS) and macrophage activation syndrome (MAS), because in patients supected of having sepsis Tocilizumab was not offered. Please modify the sentence
- Materials and Methods, line 92: Why did you use elevated IL-6 as a diagnostic criteria of MAS instead of soluble CD25 (i.e., soluble IL-2 receptor) ≥ 2,400 U/ml? Furthermore you did not indicate the upper normal limit of IL-6.
- Materials and Methods, line 98: Can you better describe the criteria to define CRS: the definition seems not to be very accurate (clinical -hyperthermia, general weakness, myalgia- and laboratory markers-C-reactive protein, ferritin, and IL-6: all criteria has to be present? Are there cut off of the laboratory markers? In addition reference 15 is a russian paper and reference 16 is not very appropriate
- Materials and Methods, line 102: Were laboratory tests described in table 1 performed at the admission? Please specify in the methods
- Results, line 133: Compared to the MAS and sepsis subtypes, clinical symptoms in CRS were less pronounced, with lower body temperature and without hepato- or splenomegaly. Can you show in table 1 the main symptoms of each cytokine storm syndrome subtypes, to support your sentence? How did you investigate the presence of hepato- or splenomegaly? Did all patients undergo to an abdominal ultrasound ? It is not very easy in COVID patients! Please, specify in the methods and describe in table 1 the frequency of hepato- and splenomegaly in the three cytokine storm syndrome subtypes
- Table 1: Can you decribe in the footnootes of table 1 laboratory reference ranges?
- Results, line 137: Most importantly, mortality in CRS patients was 14.4% compared to 67.9% and 50% in MAS and sepsis, respectively. Please specify wether the differeces were statistically significant among the three CRS subtypes.
- Results, line 147: Patients with MAS demonstrated severe hyperferritinemia, hypofibrinogenemia, and pancytopenia i.e. anaemia, leukopenia, and lymphocytopenia: these features represent the diagnostic criteria of MAS; for this reason this sentence is redundant.
- You should also evaluate the differences in the clinical effects of tocilizumab between patients affected by CRS with ARDS and those affected by MAS, who are all affected by ARDS (results, line 145). Could the differences in severity of patients influence the effectiveness of Tocilizumab? PaO2/FiO2 is associated with the risk of intubation and death (table 3)
- Among all patients treated with tocilizumab, no statistically significant differences were found between ferritin and Il-6 levels in the group of intubated vs. non-intubated patients (p=0.095, p=0.309, respectively) as well as in the deceased vs. the survivors 164 (p=0.074, p=0.114, respectively). Can you show these results in a new table? Can you also evaluate the differences in PCR considering that IDSA guidelines suggest tocilizumab in patients affected by severe or critical COVID 19 with elevated PCR (≥ 75mg/L). Can you show this result in the new table? Can you also evaluate ferritin, IL6 and PCR among all patients treated with tocilizumab and affected by ARDS subdivided in the group of intubated vs. non-intubated and in the deceased vs. the survivors ? Could the patient affected by ARDS with elevated IL-6 levels be the candidate of therapy with Tocilizumab (both features are present in your MAS patients)?
- Results, line 169: From all the biomarkers studied (Table 3) multivariate logistic regression analysis revealed few that are independent risk factors of intubation and death (Table 4). Can you better describle the results showed in table 3 and 4 ( For example an increase of dimer concentration more than…. and a decrease of lymphocite count less than …..is an indipent risk factors ofdeath)
- In the discussion, line 225: we observed the development of sepsis in two patients after treatment with this medication. Can you describe in a table the adverse drug reactions occurred in patients treated with Tocilizumab? In this way you support your sentence
MINOR COMMENTS
- Intoduction, line 55: IL-6 instead of Il-6
- Introducion, line 61. You can add this concept: “Such discrepancies may arise from heterogeneous individual responses, which likely depend on the specific pathogenic weight of IL-6 in the complex scenario of the SARS-CoV-2-elicited cytokine storm” (Pelaia C, Calabrese C, Garofalo E, Bruni A, Vatrella A, Pelaia G. Therapeutic Role of Tocilizumab in SARS-CoV-2-Induced Cytokine Storm: Rationale and Current Evidence. Int J Mol Sci. 2021 Mar 17;22(6):3059. doi: 10.3390/ijms22063059. PMID: 33802761; PMCID: PMC8002419.)
- Materials and Methods, line 68 Inclusion criteria:
1.confirmed SARS-CoV-2 infection (positive RT PCR test),
2. presence of bilateral interstitial pneumonia on a CT scan, and
3. respiratory failure with arterial partial pressure of oxygen < 60 mm Hg on room air.
- Can you rename TABLE 1? It does not comprise only laboratory features of cytokine storm syndrome subtypes
- In the discussion, line 186 “According to Ruan et al. [23], the average concentration of IL-6 (11.4 ± 8.5 mg/ml) in deceased patients is significantly higher (p<0.001) than that of 187survivors (6.8 ± 3.6 mg/ml). ng/ml instead of mg/ml
Author Response
MAJOR COMMENTS
- Corrected to „MAS and CRS”, line 22.
- IL-6 is a pleotropic cytokine produced in the early stages of inflammation and is central in driving the acute-phase response. The increased level of IL-6 correlates with MAS activity. Many authors consider this index to be one of the main ones by which one can judge the presence and activity of MAS. The upper normal limit of IL-6 is 4,0 pg/ml (this value has been added in footnote of table 1). It should be emphasized that most of the analyses of CRS in patients with COVID-19 refer to the concentration of Il-6, therefore it has become routinely measured in clinical practice, in contrast to the CD25 soluble. As our work is retrospective and CD25 soluble is not routinely measured in COVID-19 patients, it was not possible to include its level as a MAS criterion.
- Unfortunately, no official definition of CRS has been established to date, unlike, for example, HLH - the diagnosis of which is based on very specific set of criteria. Therefore, we cannot indicate cut-off points for individual inflammatory parameters. We have updated the literature to be more relevant. We have replaced items 15 and 16 with more adequate ones - currently reference 17 and 18 - line 109.
- The information was added, line 111-112.
- Table 1 has been supplemented with new information. Hepatho- and splenomegaly was detected during ultrasound examination. This is a routine test performed in ICU patients and hence there was no problem obtaining such data retrospectively. The information was added, line 111.
- Laboratory reference ranges were added in the footnootes of table 1.
- Changed to: “Most importantly, mortality in CRS patients (14.4%) was significantly (p<0.001) lower compared to MAS and sepsis (67.9% and 50% respectively). There was no difference in mortality between MAS and sepsis groups (p=0.778).”, lines 175-178.
- The sentence was delated.
- We completed the data, lines 189-196.
- We put required information in Table 3.
The analysis of efficacy of tocilizumab treatment depending on the severity of inflammatory parameters was not the goal of our study. Our study has not revealed any statistical significance between tested groups in terms of CRP and Il-6. However in all cases these factors were elevated, which in our opinion indicates a limited value of these factors as qualifiers for therapy.
We calculated and added: “A single factor logistic regression analysis was also done for all patients with ARDS and treated with tocilizumab. There were no statistically significant differences between intubated vs. non-intubated and deceased vs. survived for any investigated laboratory parameters.”, lines: 205-208. - Data from a single-factor logistic regression analysis of the risk factors for intubation and death in Table 2 (formerly Table 3) apply to all patients with CRS and MAS treated and not treated with tocilizumab. As the analysis of the same factors among patients treated with tocilizumab did not show statistically significant differences between the CRS and MAS groups, in our opinion, defining the cut-off points seems pointless and does not contribute to the work in terms of its purpose.
For the Table 3 (formerly Table 4) which contains data from multivariate logistic model analysis the critical concentration level for the independent variable cannot be evaluated (the cut-off depends on the levels of other variables of the model).
- We added table 5.
MINOR COMMENTS
13. Corrected
14. We added reference no 14, line 68.
15. Corrected
16. Corrected: “Baseline clinical characteristics and laboratory features of cytokine storm syndrome subtypes, median (QI–QIII).”
17. Corrected
Reviewer 2 Report
Authors presented very important study concerning tocilizumab therapy in COVID-19 cytokine storm syndrome. However, are some points for correction:
- In Introduction you can cite the following article https://journals.tmkarpinski.com/index.php/ejbr/article/view/320
- Data for study was only from Kyiv City Clinical Hospital. Article is cooperation between Polish and Ukrainian Authors. It is a pity that the data from the Polish hospital/s were not taken into account also.
- In Materials and Methods should be more details. It is lack of description of detection of ferritin, interleukin-6, D-Dimer, fibrinogen, calcitonin, and C-reactive protein. Should be add methods, used reagents, kit, names of producers, used devices, etc.
Author Response
- In our opinion, the article, although interesting, refers primarily to an alternative therapy to tocilizumab, which is not the topic of our work. Hence, after completing the literature in accordance with the comments of other reviewers, we decided not to cite it.
-
The study is a result of collaboration between Polish and Ukrainian researchers. However, as it is a retrospective study on the patients hospitalised in ICUs, it was not possible to compare with the Polish data, due to significant deficits in test results on Polish side. The study is an incentive to create a prospective multi-centre work.
-
Added: Respiratory index PaO2/FiO2 was determined according to the method of P. Marino using a BGA 101 gas analyzer made by Wondfo, China. C-reactive protein was determined by enzyme immunoassay using a Humastar 600 analyzer (Germany) and a reagent kit HUMAN (Germany). Procalcitonin was determined by the enzyme immunoassay using an immunological VIDAS analyzer with accessories 4700023; kit for the enzyme immunoassay of procalcitonin in human plasma and serum BRAHMS VIDAS® BRAHMS. The quantitative content of fibrinogen was determined using the Clauss method using a Siemens BFT II semi-automatic two-channel coagulometer (Siemens Healthcare Diagnostics, Germany) and a Multifibren U reagent kit (Germany). To determine the D-dimer an enzyme-linked immunosorbent assay was used; Humastar 600 analyzer and a HUMAN reagent kit. To determine interleukin-6 a set of reagents for the enzyme-linked immunosorbent assay for determination of the concentration of interleukin-6 in blood serum and urine Human Interleukin 6 was used, IL-6 ELISA KIT (Germany). The determination was carried out on an automatic ELISA analyzer EVOLIS Twin Plus, Bio-Rad, France. Ferritin was determined by enzyme immunoassay using a Humastar 600 analyzer (Human GmbH, Germany) and a reagent kit HUMAN (Germany).”, lines: 113-128.
Reviewer 3 Report
Oliynyk et al., studied the “Efficacy of Tocilizumab Therapy in Different Subtypes of COVID-19 Cytokine Storm Syndrome”.
The authors have performed a retrospective analysis of COVID-19 patients treated with or without anti-IL-6 antibody, i.e., tocilizumab. In a nutshell, they have found that tocilizumab is not effective in COVID-19 patients with cytokine releasing syndrome; somewhat, it has enhanced mortality. Besides, COVID-19 patients with macrophage activation syndrome treated with or without tocilizumab have improved mortality and intubation rate.
The inclusion and exclusion criteria are good, and the methods used for the statistical analysis are apt.
However, the following corrections and suggestions should be made to accept the manuscript for publication.
Expand the terms COVID-19, SARS-CoV-2, RR, CI, HLH, CAR T cells, SOFA, and others at first instance.
Line 36-40: This broad reference would better suit here. Pathophysiology of COVID-19, including cytokine storm was explained. https://doi.org/10.1016/j.nantod.2020.101051
Line 57: “blocking cytokine signalling may impair clearance of the SARS-CoV-2” the sentence is wrong.
Blocking cytokine signalling may assist the clearance of the SARS-CoV-2. Mainly, the hypothesis is blocking signalling would reduce the clinical symptoms, which sequentially stabilise the hyperactivated immune response.
Line 58: “prescribed to patients with severe COVID-19” ------ It has been prescribed to severe as well as mild COVID-19 conditions.
Line 60-62: Font should be changed.
Line 103: “procalcitonin (PCT)” should be expanded earlier, not here.
Line 137-139: This sentence should be placed after line 149.
Table 2 should be represented as a bar graph; for better visibility.
Line 183: p <0.001 ---- p<0.001
Line 206-210: These two recent references should be discussed. https://doi.org/10.1056/NEJMoa2028700; https://doi.org/10.1056/NEJMoa2100433
Author Response
- We expanded all the terms.
- We added the reference (no 3, line 42)
- Corrected, line 61.
- This is true, that is why we used “is mainly prescribed”
- Changed
- Changed
- Changed, lines 175-178.
- Table 2 has been replaced by a bar graph
- Changed
- We added references (no 30 and 31) and commented, lines: 254-272.
Round 2
Reviewer 1 Report
The work can be accepted in its present formthe work can be accepted in its present form
Reviewer 2 Report
Authors significantly corrected manuscript according to reviewer's suggestions. Recently, I recommend article for publication.